# The Effects of a Floss Band on Ankle Range of Motion, Balance, and Gait in Chronic Stroke: A Randomized Controlled Study

**DOI:** 10.3390/healthcare12232384

**Published:** 2024-11-27

**Authors:** Byoung-Hyoun Moon, Ji-Won Kim

**Affiliations:** Department of Physical Therapy, Nambu University, 23, Cheomdanjungang-ro, Gwangsan-gu, Gwangju 62271, Republic of Korea; mbh930@naver.com

**Keywords:** ankle, balance, chronic stroke patients, floss band, gait, range of motion

## Abstract

**Background/Objectives**: Stroke patients generally have balance and gait dysfunction due to decreased range of motion (ROM) and muscle strength of the ankle joint. A therapist can apply a floss band to enhance ROM, pain control, jump performance, strength, myofascial release, and recovery from fatigue. This study compared the immediate effects of floss band application on ankle ROM, balance, and gait ability in stroke patients. **Methods**: This study randomized 40 chronic stroke patients to either the floss (n = 20) or sham (n = 20) band group. The intervention consisted of wrapping the ankle with a band (floss and sham). Balance and gait ability were measured using APDM Mobility Lab system equipment. Outcome measures were assessed at baseline and immediately after applying the floss or sham band. **Results**: There were significant improvements in dorsiflexion (DF), WBLT, static balance, and foot strike in those who used the floss band compared to the sham group (all *p* < 0.05). **Conclusions**: The floss band improved ankle DF, balance, and gait, indicating that it is a feasible therapeutic method for improving ankle DF, balance, and gait in chronic stroke patients.

## 1. Introduction

More than 50% of stroke patients experience gait problems and have a reduced likelihood of regaining their previous motor skills [1]. Limited ankle dorsiflexion (DF) in stroke patients is one of the main causes of functional limitations after stroke [2,3]. The causes of limited DF are spastic plantar flexors and weakened dorsiflexors, which make it difficult for stroke patients to perform active DF [4]. Limited DF in stroke patients can restrict fascia, ligaments, and muscles, resulting in a loss of normal joint range of motion (ROM) and other accessory movements of the ankle joint and adjacent joints [5,6]. Insufficient DF causes asymmetric weight bearing on the affected leg, which reduces the balance and gait of stroke patients and ultimately interferes with their ability to perform activities of daily living (ADL) or participate socially [7].

Regaining efficient gait is a major goal for stroke survivors [8]. Therefore, to overcome mobility impairment in stroke patients, interventions that improve ankle DF are commonly used during rehabilitation, including calf muscle stretching, ankle foot orthoses, ankle ROM exercises, robot training, and functional electrical stimulation of the dorsiflexors [9,10,11,12,13]. Recently, there has been increased research on the effectiveness of the floss band at improving DF. A floss band is a simple intervention that involves wrapping a joint or muscle with a thick elastic band to cause vascular occlusion and then performing active exercises for 1~3 min [14]. The mechanism of the floss band technique includes fascial shearing and occluding blood flow to the muscle [15]. Driller and Overmayer (2017) reported that using a floss band on either ankle increased DF, plantarflexion, and jump performance in 52 recreational athletes [16]. Vogrin et al. (2020b) found that the floss band technique improved ROM and muscle function for 45 min [17]. In addition, athletes who used the floss band technique exhibited improvement in DF in a weight-bearing posture that was sustained for 72 h [18].

Previous studies have consistently demonstrated the potential of floss band application on the ankle to improve ROM, muscle function, and jump performance in athletes. In stroke patients, ankle ROM is a critical factor for effective gait performance [7]. Park and Hwang (2016) reported that applying myofascial release techniques to the lower extremities of stroke patients significantly improved balance and gait abilities [19]. However, no studies have investigated the effects of ankle floss band application for myofascial release on ROM, balance, and gait specifically in stroke patients. Therefore, this study analyzed the effects of a floss band applied to the ankle joint in chronic stroke patients on ROM, static balance, and gait ability and compared these outcomes with those of a sham group. The primary hypothesis of this study was that the application of a floss band to the ankle would immediately improve ankle dorsiflexion range of motion (DF ROM), balance, and gait performance in chronic stroke patients. The secondary hypothesis was that the therapeutic effects of the floss band, compared to a sham band, would result in greater improvements in ankle ROM, balance, and gait measures.

## 2. Materials and Methods

### 2.1. Participants

This study design used a randomized controlled design (with a control group) to investigate the effects of 2 min of low-intensity exercise using a floss band on chronic stroke patients. (Figure 1). This study was conducted between December 2021 and July 2022. G*Power software (ver. 3.1 for Windows; Franz Faul, University of Kiel, Kiel, Germany) was used to calculate the sample size needed, assuming a medium effect size of 0.25 for repeated measures, with both within- and between-group analyses, an alpha error probability of 0.05, and a power of 0.80. A total sample size of 34 subjects was estimated. We enrolled 40 people, assuming a dropout rate exceeding 10%. The effect size was set to 0.25, which is a medium effect size based on previous study [20]. The subjects were recruited from H Rehabilitation Hospital, the Republic of Korea. All the subjects were diagnosed with ischemic or hemorrhagic stroke, as confirmed by computed tomography or magnetic resonance imaging. Based on our study screening criteria, we enrolled chronic patients 6 months after stroke onset who were able to walk 10 m independently, had the ability to follow simple instructions, had weakness in dorsiflexing the ankle on the affected side, had a modified Ashworth scale (MAS) score of <G2 for the ankle joint, had an adequate cognitive status—taken as a Mini-Mental State Examination (MMSE) score of ≥24—and had no orthopedic problems involving the lower extremities that would affect gait. The exclusion criteria were impaired lower extremity function due to other causes; dizziness, hemianopia, or other symptoms indicating vestibular dysfunction; serious heart disease or the use of a pacemaker; and inability to tolerate the floss band intervention. This study was approved by the Ethics Committee of Nambu University (1041478-2021-HR-022) and the clinical trial registry (KCT0009927). This study was performed in accordance with the Declaration of Helsinki. The patients provided informed consent before participating in this study. All the participants voluntarily agreed to participate after fully understanding the objectives and procedures used in this study.

### 2.2. Procedures

This was a cross-sectional study. The clinical information collected included age, height, weight, gender, affected side, stroke type, Korean Mini-Mental State examination (K-MMSE) score, MAS, Brunnstrom recovery stage (BRS), Berg Balance Scale (BBS) score, modified Barthel index (MBI), and timed up and go (TUG) performance. The patients were selected using a randomization program (www.randomizer.org) and randomly assigned to the floss band (n = 20) or sham (n = 20) group. The participants visited a laboratory for a single testing session. The participants performed an ankle ROM test, weight-bearing lunge test (WBLT), standing balance test, and gait ability test before and after the application of a floss band or sham band. The floss bands were applied by a qualified physiotherapist. After applying the floss band or sham band, the participants performed low-intensity active exercises involving ankle DF and plantar flexion for 2 min.

### 2.3. Interventions

All the patients received conventional treatment, which included functional electric stimulation (FES), joint mobilization, passive stretching, balance training, and gait training. Floss bands were applied by a qualified therapist with at least 5 years of experience. To prevent potential assessment bias, the assessors were blinded to this study’s goal and group allocation when evaluating the outcome measures. The outcome measures were evaluated before the intervention and immediately after the intervention with the floss band.

#### 2.3.1. Floss Band Intervention

The floss band intervention used the standard floss band technique, wrapping a floss band (lime green Sanctband Comprefloss™, 2” × 3.5 m; PENTEL, Shah Alam, Malaysia) made of natural rubber tightly around the ankle on the affected side. The floss band began at the fifth metatarsal and went horizontally around the metatarsals twice, in a figure eight to the medial malleolus, over the Achilles tendon and the lateral malleolus three times, and around the medial malleolus again, before passing twice from the medial malleolus over the Achilles tendon to the lateral malleolus and forming an end knot (Figure 2) [16]. The participants were instructed to perform low-intensity active exercises involving ankle DF and plantar flexion for 2 min after applying the floss band (Figure 3). Then, the floss band was removed, and the patient was asked to walk lightly on level ground for about 1 min to allow reperfusion to normalize blood flow.

#### 2.3.2. Sham Floss Band Intervention

The wrap used in the sham group lacked the elasticity of a floss band; the ankle was wrapped in the same manner, but loosely to facilitate blood circulation. Similarly, the participants were instructed to perform low-intensity active exercises involving ankle DF and plantar flexion for 2 min after applying the sham floss band. Finally, the sham floss band was removed, and the patient was asked to walk lightly on level ground for about 1 min under the same conditions as in the floss band group.

### 2.4. Outcome Measurements

The participants were assessed before and immediately after the band intervention (floss or sham), including in terms of ankle passive ROM, weight-bearing lunge test (WBLT) performance, static balance, and gait ability.

#### 2.4.1. Ankle Passive Range of Motion

Ankle passive ROM was measured using a universal goniometer in a non-weight-bearing position. The subjects were positioned prone with the knee joint at 90° [21]. The ankle joint was positioned at 0° of eversion and inversion. The goniometer axis was placed beneath the lateral malleolus, and the stationary arm was positioned parallel to the fibula. The movable arm was positioned parallel to the fifth metatarsal, with the ankle in a neutral position. The measurement was repeated three times, and the average value was calculated. The intrarater reliability of the passive ankle DF had an intraclass correlation coefficient (ICC) of 0.92–0.96 [22,23].

#### 2.4.2. Weight-Bearing Lunge Test

The WBLT was performed to assess DF in a functional ankle joint. A measuring tape was placed horizontally on the floor perpendicular to a wall. The participants placed their affected-side foot on the tape with their big toe contacting the wall and were instructed to touch the wall with the knee on the affected side. While maintaining this position, they were instructed to perform lunges by bending their knee, aiming for contact between their knee and the wall while keeping their heel firmly fixed on the floor. Once they were able to maintain knee and heel contact, the affected side foot was moved away from the wall, and they repeated the lunge test. The test was performed with 1 cm increases until knee and heel contact were no longer maintained. The maximum lunge distance was the farthest distance from the wall to the big toe with the foot staying on the floor (without the heel lifting) when the knee touched the wall. Three practice trials of the WBLT were conducted, followed by three test trials. For data analysis, the average of the three trials was calculated. The intrarater reliability of the WBLT was high (ICC = 0.98–0.99) [24].

#### 2.4.3. Static Balance Ability

Static balance was assessed using the APDM Mobility Lab™ Opal inertial sensor system (APDM, Portland, OR, USA). The test was conducted in a quiet treatment room. During the test, the participants were barefoot and wore three Opal inertial sensors: one over their clothing at the level of the fifth lumbar vertebra and one on each ankle. Each subject was instructed to maintain their balance as stably as possible in a barefoot standing position (10 cm between heels, toe-out of 5°) for 30 s. The test was repeated three times at 30-s intervals. The static balance outcome measure was the postural sway area (cm/s^2^). The signal was sampled, processed automatically, and streamed to a laptop using Mobility Lab™ software (v1.0, Mobility Lab, Arlington, VA, USA). Balance assessed with the APDM had a high ICC or 0.60–0.89 [25].

#### 2.4.4. Gait Ability

The APDM Mobility Lab™ Opal inertial sensor system (APDM) was used to assess gait based on foot strike (FS) and toe-off (TO) angles. Data were collected from the sensor wirelessly at a sampling rate of 128 Hz and processed to quantify postural sway parameters. The test was conducted in a quiet treatment room. During the test, the participants were barefoot and wore three Opal inertial sensors: one over their clothing at the level of the fifth lumbar vertebra and at each ankle. Verbal instructions were given to ensure accuracy. The subject was told to stand still at the start line until the first long tone was heard, at which time they started walking at a comfortable natural pace. When a second tone was heard after 2 min, the participants were asked to stop walking. After practicing for 30 s to become familiar with the test, the participants were asked to walk back and forth along a straight 10-m corridor at their usual pace for 2 min without a walking assist. All procedures were performed by an experienced physical therapist who stood nearby and ensured the participants’ safety. Gait parameters extracted for analysis included the FS and TO angles. Signals were streamed to a laptop for automatic processing using Mobility Lab™ software. The gait measured by the APDM device had excellent reliability, with an ICC of 0.905–0.991 [26].

### 2.5. Statistical Analysis

The data were analyzed using IBM SPSS Statistics 22.0 (Chicago, IL, USA). Descriptive statistics (mean ± standard deviation) were calculated for age, height, weight, K-MMSE, MAS, BRS, BBS, MBI, and TUG. Frequency analysis was used to assess gender, affected side, and stroke type. Normality was tested using the Kolmogorov–Smirnov test. An independent *t*-test was used to make comparisons between groups (floss vs. sham), and a paired *t*-test was performed to compare times (before vs. after). Statistical significance was set at *p* < 0.05. Cohen’s d effect sizes were calculated to aid in interpreting the results: 0.00 to 0.19 were considered trivial effect sizes, 0.20 to 0.49 small effect sizes, 0.50 to 0.79 moderate effect sizes, and above 0.80 large effect sizes.

## 3. Results

We enrolled 40 stroke patients (22 men and 18 women). There were no serious adverse events or patient withdrawals.

### 3.1. Baseline

Table 1 summarizes the patients’ demographics and stroke characteristics. There were no significant baseline differences between groups in terms of age, height, weight, gender, affected side, stroke type, K-MMSE, MAS, BRS, BBS, MBI, or TUG.

### 3.2. Passive ROM

Stroke patient intervention with the floss band had a significantly greater improvement in passive DF ROM than those treated with the sham floss band (*p* = 0.000, effect size = 2.58). However, the passive ROM of plantar flexion showed no significant differences (*p* = 0.165, effect size = 0.45) (Table 2).

### 3.3. WBLT

The WBLT performance improved after the floss band intervention compared to before the intervention, and the floss band led to significantly greater improvement in the WBLT than the sham floss band (*p* = 0.006, effect size = 0.92) (Table 2).

### 3.4. Balance Ability

The sway area of static balance improved after the floss band intervention compared to before the intervention, and the floss band led to significantly greater improvement in static balance than the sham floss band (*p* = 0.008, effect size = 0.89) (Table 2).

### 3.5. Gait Ability

FS improved after the floss band intervention compared to before the intervention, and the floss band led to significantly greater improvement in FS than the sham floss band (*p* = 0.045, effect size = 0.65) (Table 2). However, TO showed no significant differences (*p* = 0.997, effect size = 0.001) (Table 2).

## 4. Discussion

This study is the first to investigate the effects of applying floss bands to the ankle joint on DF and plantarflexion, functional joint ROM under weight-bearing conditions, static balance, and gait ability in stroke patients. We found improvements in DF, static balance, and gait ability when floss bands were applied to the ankle joint. These significant effects of the floss band may have practical implications for ADL in stroke patients, given the improvements in balance and gait abilities.

Applying floss bands to the ankle joint in stroke patients significantly improved DF compared to the sham band. This is consistent with the findings of Driller and Overmayer (2017), who reported that floss bands improved ankle DF in athletes and recommended the application of floss bands before sports events to increase ankle DF and help prevent injury. Stroke patients frequently exhibit restricted dorsiflexion (DF) compared to healthy individuals, primarily due to intramuscular stiffness. This stiffness can disrupt both intramuscular and extramuscular support structures, ultimately contributing to functional limitations [27,28]. The physiological mechanism of floss bands involves compressing the muscles, stimulating the nervous system to separate the fascia, and allowing the fascia to move freely [29]. Previous studies have reported that using myofascial release techniques in stroke patients improves balance and gait abilities [19]. Stroke often involves paralysis around the ankle, limiting ankle ROM, causing muscle weakness, and restricting sliding of the fascia. Therefore, applying floss bands to stroke patients may improve restricted DF by promoting myofascial separation through muscle compression.

Our results also showed that applying floss bands in stroke patients improved WBLT performance significantly. The WBLT is used to assess ankle DF, flexibility, stability, and balance in a closed kinetic chain. One plausible explanation is that the floss band strengthens the ankle muscles, which in turn improves functional ankle DF. Stevenson et al. (2019) achieved improvements in WBLT performance using a floss band at the ankle, suggesting that the increased functional ROM facilitates treatment [30]. Although the mechanism of floss bands remains known, Vogrin et al. (2020a) showed that blood flow restriction leads to the accumulation of metabolites, synthesis of muscle proteins, and increase in growth hormones [31]. The subsequent reperfusion normalizes the supply of nutrients and hyaluronic acid to local areas, increasing ROM [32,33]. We postulate that the floss band increased WBLT performance via this mechanism.

In our subjects, the sway area of static balance was reduced significantly more with the floss band compared to the sham floss band. The static balance of stroke patients is related to ankle muscle strength [34], flexibility [35], and proprioception [36]. Kim and Kim (2018) reported that static balance when standing was correlated with ankle ROM and lower extremity muscle strength [37]. We thus postulate that the application of a floss band affected balance by increasing the ankle ROM. Chang et al. (2021) reported that applying a floss band to the knee in men improved lower extremity flexibility, quadricep strength, and balance [38]. The application of a floss band relaxes soft tissues and increases muscle elasticity, thereby reducing joint stiffness and improving balance. Therefore, the application of a floss band in stroke patients can improve static balance while standing by improving ankle ROM, muscle strength around the ankle, and coordination on the affected side.

We observed a significantly greater improvement in FS during gait with the floss band compared to the sham floss band. FS refers to the initial contact during the gait cycle and is closely related to the range of DF [39]. Ravichandran and Janakiraman (2021) claimed that improving the functional range of DF is essential for post-stroke rehabilitation because limited ankle DF is common in most chronic stroke patients and is related to balance and gait. Stroke patients often have limited ankle DF, and abnormal initial contact occurs at the forefoot or outer edge of the foot, negatively affecting the stability of the loading response phase [40]. Kaneda et al. (2020b) reported that applying a floss band to the gastrocnemius improved flexibility, the rate of force development, and DF [41]. It is thought that the application of a floss band increased DF by reducing the tension in the calf muscle and surrounding soft tissues, thereby increasing the FS angle. Stroke results in complex sensorimotor deficits, including muscle weakness, impaired muscle control, spasticity, and proprioceptive deficits that affect balance and gait. Cho et al. (2021) reported that ankle muscle contraction and active–resistive strength training in chronic stroke improved ankle muscle strength, proprioception, balance, and gait [34]. Similar to previous authors, we think that the application of a floss band increased DF by relaxing the calf muscle and surrounding soft tissues, thereby increasing the FS angle. Since the application of a floss band enhances DF and balance, it should help to improve the gait of stroke patients.

We observed a significant effect of the floss band on plantar flexion and TO angle in stroke patients. We postulate that there was no difference in plantar flexion between the groups in this study because the plantar flexion angle of the stroke patients was within the normal ROM. TO refers to the initial swing during the gait cycle and is closely related to plantar flexion [42].

This study has several limitations. First, it analyzed the immediate effect of the floss band, and further research needs to examine changes over time. Second, we applied a floss band only to the ankle of stroke patients; further studies need to examine the effects on other joints in these patients. Third, this study did not compare our invention with other common interventions in stroke patients. Lastly, our results may not be generalizable to all patients with stroke as we studied stroke patients in Brunnstrom stage 4 or 5.

## 5. Conclusions

This study investigated the effects of a floss band on the ankle ROM, WBLT performance, balance, and gait of stroke patients. The floss band group had significantly improved DF, FS, and balance compared to the sham group. Therefore, this study recommends the use of floss band interventions in clinical practice as a feasible treatment method for improving ankle DF, balance, and gait in chronic stroke patients. The study findings serve as a valuable reference that could aid the clinical application of floss bands.

## Figures and Tables

**Figure 1 healthcare-12-02384-f001:**
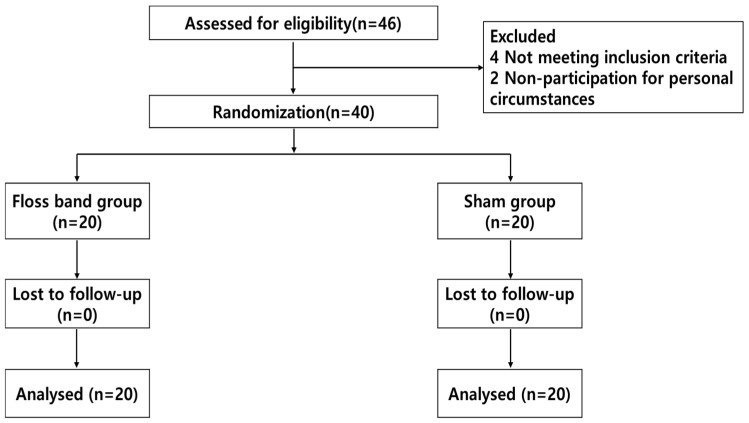
Flow chart of this study.

**Figure 2 healthcare-12-02384-f002:**
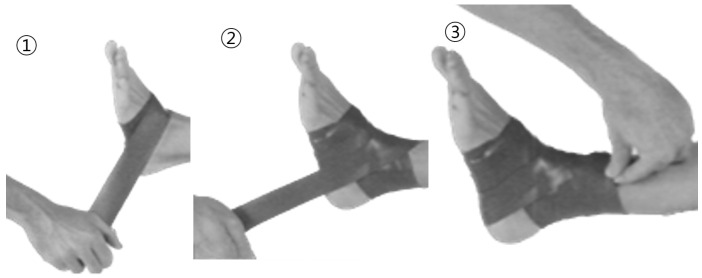
Floss band technique in stroke patients.

**Figure 3 healthcare-12-02384-f003:**
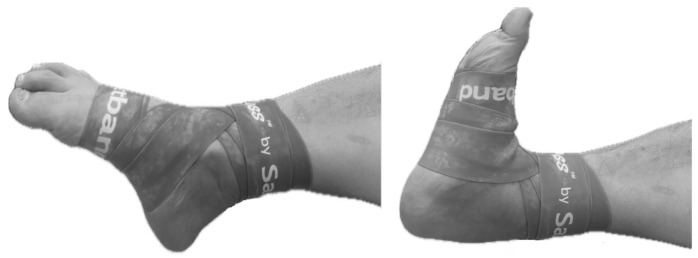
Low-intensity active exercise (ankle dorsiflexion and plantar flexion).

**Table 1 healthcare-12-02384-t001:** Demographic and baseline characteristics of participants.

	Floss Band Group (n = 20)			Sham Group (n = 20)			t/χ^2^	*p*
Age (years)	58.15	±	11.93	60.4	±	10.29	−579	0.566
Height (cm)	164.8	±	8.08	167.4	±	8.28	−1.16	0.253
Weight (kg)	64.34	±	11.02	66.4	±	8.56	−175	0.862
Gender (male/female)	10	/	10	12	/	8	0.241	0.623
Affected side (R/L)	9	/	11	6	/	14	1.336	0.248
Stroke type (hemorrhage/ischemia)	5	/	15	9	/	11	2.046	0.153
K-MMSE	25.65	±	2.11	24.75	±	2.47	1.136	0.263
MAS	0.8	±	0.5	0.53	±	0.54	1.757	0.087
BRS	4.6	±	0.48	04.4	±	0.50	1.087	0.284
BBS	42.15	±	6.24	39.85	±	9.28	0.721	0.475
MBI	63.55	±	9.53	60.1	±	14.23	0.756	0.454
TUG	10.16	±	1.07	10.18	±	1.18	0.011	0.991

R: right, L: left, K-MMSE: Korean Mini-Mental State Examination, MAS: modified Ashworth scale, BRS: Brunnstrom recovery stage, BBS: Berg Balance Scale, MBI: modified Barthel index, TUG: timed up and go test. Values are presented as mean ± standard deviation.

**Table 2 healthcare-12-02384-t002:** Pre- and post-intervention measures for floss band and Sham group trials.

		Pre			Post			t	*p*
DF (°)	Sham	3.10	±	2.38	3.40	±	2.16	−0.90	0.379
	Floss	2.60	±	1.47	8.90	±	2.10 ^†^	−14.26	0.000 *
PF (°)	Sham	52.90	±	1.94	53.10	±	1.62	−0.62	0.541
	Floss	53.85	±	1.95	54.00	±	2.34	−0.30	0.772
WBLT (cm)	Sham	7.65	±	1.84	7.90	±	1.80	−1.75	0.096
	Floss	8.65	±	1.50	9.40	±	1.43 ^†^	−3.94	0.001 *
SB (mm^2^)	Sham	7.39	±	3.87	7.25	±	3.47	0.385	0.704
	Floss	7.00	±	5.25	4.31	±	3.16 ^†^	3.84	0.001 *
FS (°)	Sham	9.38	±	6.47	9.58	±	5.74	−0.41	0.687
	Floss	10.20	±	4.16	13.09	±	4.95 ^†^	−4.88	0.000 *
TO (°)	Sham	14.96	±	3.91	15.04	±	4.18	−0.20	0.843
	Floss	15.74	±	5.81	15.04	±	4.16	0.75	0.463

Values are presented as mean ± standard deviation. Paired *t*-test: * *p* < 0.05, independent *t*-test: ^†^ *p* < 0.05. DF: dorsi flexion, PF: plantar flexion, WBLT: weight-bearing lunge test, SB: static balance, FS: foot strike, TO: toe-off.

## Data Availability

The original contributions presented in this study are included in this article; further inquiries can be directed to the corresponding author.

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
