# Peer review of "The Effects of a Floss Band on Ankle Range of Motion, Balance, and Gait in Chronic Stroke: A Randomized Controlled Study"

_healthcare, 2024, doi:10.3390/healthcare12232384_

Round 1
Reviewer 1 Report
Comments and Suggestions for Authors
This study is a study on the effects of floss band on the range of joint motion, balance, and gait of chronic stroke patients, and it can attract readers' attention because it confirmed the effects of floss band outside of general rehabilitation training. Please consider the following and revise it.
1. The introduction of the research method in the Abstract is insufficient. Please add more information about the exercise method.
2. Regaining efficient gait is a major goal for stroke survivors. -> Please add references for all the sentences you provided.
3. Please provide additional explanations about the variables that determine sample size.
4. Please describe the research ethics in detail.
5. A description of conventional treatment is needed. Please provide detailed explanations about whether it was performed by one person, whether it was blinded, and how many minutes it was performed.
6. If there are references about floss bands, please provide them. Also, please provide separate pictures of the application methods in the order of 1, 2, 3, and 4. Also, it would be easier to understand if there were pictures of low-intensity active exercise.
7. A detailed explanation of the research procedure is needed. What kind of exercise was performed after band application, when did the evaluation start, what kind of evaluation was performed, how many times was it performed, by whom, etc.
8. Please add an explanation of the effect size in the statistical method.
9. Since the immediate effect was confirmed, independent t and paired t seem more appropriate. Please think about it and revise it.
10. Please recommend the clinical effect or intervention of floss band for stroke patients.
Author Response
Comments 1:
- The introduction of the research method in the Abstract is insufficient. Please add more information about the exercise method.
Response 1: Thank you for your comments. We have added information about exercise methods in abstract.
Comments 2:
- Regaining efficient gait is a major goal for stroke survivors. -> Please add references for all the sentences you provided.
Response 2: We have added reference to sentence (reference 8).
Comments 3:
- Please provide additional explanations about the variables that determine sample size.
Response 3: We have provided additional explanations about the variables that determine sample size, referencing prior research to clarify these factors (Page 2, Line 76-77).
Comments 4:
- Please describe the research ethics in detail.
Response 4: We have included the research ethics in detail (Page 2, Line 88-92).
Comments 5:
- A description of conventional treatment is needed. Please provide detailed explanations about whether it was performed by one person, whether it was blinded, and how many minutes it was performed.
Response 5: We have added a detailed description of the conventional treatment within the interventions section. Additionally, details of the FLOSS band intervention (number of intervening physical therapists, blinding, and intervention duration) were described in manuscript (Page 3, Line 106-108, Line 110-115).
Comments 6:
- If there are references about floss bands, please provide them. Also, please provide separate pictures of the application methods in the order of 1, 2, 3, and 4. Also, it would be easier to understand if there were pictures of low-intensity active exercise.
Response 6: We have added references to prior research on floss bands to support the content (reference 16). Additionally, I updated the images to include separate pictures illustrating the application methods in the specified order (1, 2, and 3), as well as images showing low-intensity active exercises for easier understanding (Figure2-3).
Comments 7:
- A detailed explanation of the research procedure is needed. What kind of exercise was performed after band application, when did the evaluation start, what kind of evaluation was performed, how many times was it performed, by whom, etc.
Response 7: We have added a detailed explanation of the research procedure within the procedure section (Page 3, Line 95-108).
Comments 8:
- Please add an explanation of the effect size in the statistical method.
Response 8: We have added an explanation of the effect size within the statistical methods section (Page 5, Line 205-207).
Comments 9:
- Since the immediate effect was confirmed, independent t and paired t seem more appropriate. Please think about it and revise it.
Response 9: We have revised the statistical method in accordance with reviewer`s suggestion, changing to independent t-tests and paired t-tests as recommended.
Comments 10:
- Please recommend the clinical effect or intervention of floss band for stroke patients.
Response 10: We have included a recommendation on the clinical effects and potential intervention of floss bands for stroke patients in the conclusion section (Page 9, Line 320-322).
Reviewer 2 Report
Comments and Suggestions for Authors
Dear authors,
Your study explores an innovative and under-researched area: the application of floss bands in stroke rehabilitation. I would like to offer a few comments that could enhance the impact and clarity of your article:
Introduction: Clearly state your hypothesis and make a stronger case for why findings in athletic populations are relevant for stroke rehabilitation.
Methods: I suggest to provide detail participant screening criteria and explain how randomization was checked for baseline equivalency. Providing effect sizes for all measures would aid in interpreting practical significance.
Discussion: Expand on the mechanisms by which floss bands could impact stroke patients specifically and compare findings with other common stroke interventions.
Figures and Tables: Figures are clear but could be improved by indicating effect sizes where relevant. I suggest additional annotations on key findings that would also enhance clarity.
Thank you
Comments on the Quality of English Language
The manuscript is generally clear, but minor grammatical errors and phrasing issues could benefit from a final review. Could you please check it?
Author Response
Comments 1:
Introduction: Clearly state your hypothesis and make a stronger case for why findings in athletic populations are relevant for stroke rehabilitation.
Response 1: Thank you for your feedback. We have included a clear statement of the hypothesis in the final paragraph of the introduction. Additionally, we have added sentences explaining how outcomes from athletic populations can be applied to stroke rehabilitation (Page 2, Line 52-65).
Comments 2:
Methods: I suggest to provide detail participant screening criteria and explain how randomization was checked for baseline equivalency. Providing effect sizes for all measures would aid in interpreting practical significance.
Response 2:
We have added participant screening criteria (Page 2, Line 80-88).
Baseline equivalency was checked tested using the Kolmogorov–Smirnov test (Page 5, Line 202).
Additionally, effect sizes for all measures have been provided to assist in interpreting the practical significance of the results (Page 6-7, Line 222-238).
Comments 3:
Discussion: Expand on the mechanisms by which floss bands could impact stroke patients specifically and compare findings with other common stroke interventions.
Response 3: We have added a description of the mechanisms by which the floss band may affect stroke patients. Additionally, we have included a comparison of the findings with other common stroke interventions (Page 8, Line 256-265).
Comments 4:
Figures and Tables: Figures are clear but could be improved by indicating effect sizes where relevant. I suggest additional annotations on key findings that would also enhance clarity
Response 4: The table has been revised due to changes in the statistical methods. Effect sizes have been presented in the results section.
Round 2
Reviewer 1 Report
Comments and Suggestions for Authors
Thank you for your hard work in research.
It has been revised as per your comment.
Thank you.